# Fine-Grained Emotion Recognition with In-Context Learning: A Prototype Theory Approach

## Abstract

In-context learning (ICL) achieves remarkable performance in various domains such as knowledge acquisition, commonsense reasoning, and semantic understanding. However, its effectiveness deteriorates significantly in emotion detection tasks, particularly in fine-grained emotion recognition. The reasons behind this decline still remain unclear. In this paper, we explore the underlying reasons of ICL's suboptimal performance through the lens of prototype theory. Our investigation reveals that ICL aligns with the principles of prototype theory when applied to fine-grained emotion recognition tasks. According to prototype theory, effective emotion recognition requires: Referencing well-represented emotional prototypes that are similar to the query emotions, and making predictions based on the closest emotional similarity. Building on this insight, ICL has three main shortcomings: (i) It uses oversimplified single-emotion labels for prototypes, leading to inaccurate emotion representation. (ii) It references semantically similar but emotionally distant prototypes. (iii) It considers all emotion categories as candidates, leading to interference from irrelevant emotions and inaccurate predictions.

To address these shortcomings, we propose an **E**motion **C**ontext **L**earning method (E-ICL) for fine-grained emotion recognition. E-ICL first employs a dynamic soft-label strategy to create multi-dimensional emotional labels for accurate prototype representation. It then selects emotionally similar prototypes as references for emotion prediction. Finally, it uses an emotion exclusion strategy to eliminate interference from dissimilar emotions by selecting similar emotions as candidates, resulting in more robust and accurate predictions. Note that our approach is implemented with the aid of a plug-and-play emotion auxiliary model, requiring no additional training. Extensive experiments conducted on fine-grained emotion datasets—EDOS, Empathetic-Dialogues, EmpatheticIntent, and GoEmotions—demonstrate that E-ICL significantly outperforms existing methods in emotion prediction performance. Moreover, even when the emotion auxiliary model accounts for less than 10% of the LLMs' capacity, E-ICL consistently boosts LLM performance by over 4% across multiple datasets.

## 1 Introduction

Achieving human-like intelligence necessitates that machines understand and interpret nuanced human emotions. Fine-grained emotion recognition (Liew & Turtle, 2016; Abdul-Mageed & Ungar, 2017) aims to identify a wide range of subtle emotion categories in queries, making it a crucial component in various downstream tasks such as empathetic dialogue systems (Rashkin et al., 2019; Sabour et al., 2022; Li et al., 2022; 2020; Yang et al., 2023b; Zhao et al., 2022), sentiment analysis (Wang et al., 2016; Schuff et al., 2017; Guzman & Maalej, 2014), and emotional support systems (Saha et al., 2021; 2022; Peng et al., 2022; Tu et al., 2022). Earlier studies developed small-scale models to identify fine-grained emotions in given queries (Kim et al., 2021b; Majumder et al., 2020; Xie et al., 2019; Majumder et al., 2019; Ghosal et al., 2019). While these approaches were successful to some extent, they often lacked flexibility and generalizability, being limited by the emotions and knowledge contained within specific datasets. Recent advances about in-context learning (ICL) has shown promising performance across a wide range of tasks by prompting large

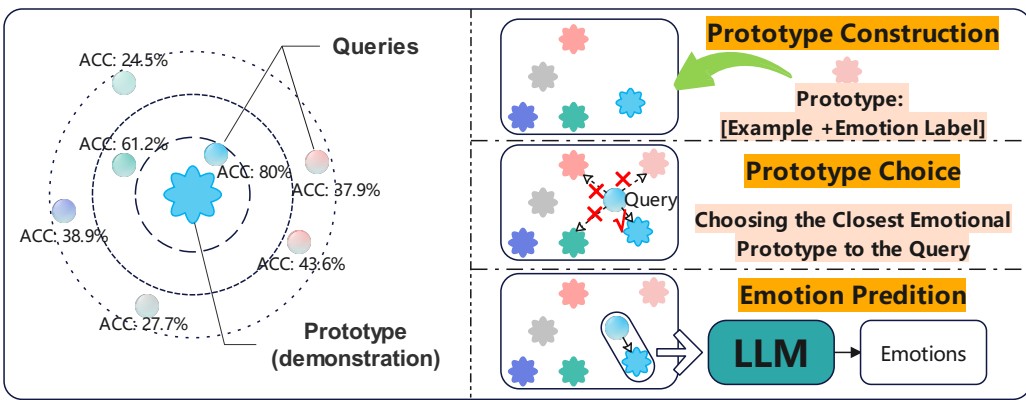

(a) Prototype Theory      (b) Key Steps of Prototype Theory

Figure 1: Illustration of prototype theory and its key steps. (a) Prototype theory: Queries closer to the prototype are more easily classified accurately. (b) Key steps: Three steps when applying prototype theory to fine-grained emotion recognition, including prototype construction, prototype selection, and emotion prediction. Moreover, for detailed definitions of key terms like demonstration and example, please refer to Section 3.

language models (LLMs) to interpret queries alongside relevant demonstrations (Rae et al., 2021; Liu et al., 2021; Yang et al., 2023a; Xiao et al., 2023; Liu et al., 2021; Rubin et al., 2021; Fu et al., 2022), exhibiting remarkable flexibility and generalizability. However, ICL still struggles with fine-grained emotion recognition with poor performance (Zhao et al., 2023; Schaaff et al., 2023; Yang et al., 2024c; Qian et al., 2023). Furthermore, its performance declines even more when the demonstrations used are less relevant (Xu et al., 2024; Liu et al., 2021). For example, randomly selected demonstrations from the training dataset perform worse than semantically relevant ones, and demonstrations outside the training dataset fare even worse.

We conduct pilot experiments to explore the reasons behind ICL's suboptimal performance in fine-grained emotion recognition (Details shown in Appendix A). We construct 9,728 samples from the emotion recognition datasets, each consisting of a query accompanied by demonstrations. We then prompt LLMs to predict fine-grained emotion categories for queries. The results reveal that queries with higher semantic similarity to the demonstrations exhibited better performance, as shown in Figure 1(a). Interpreting the demonstrations as emotion prototypes, this finding suggests that ICL aligns with prototype theory (Rosch, 1978; Kamp & Partee, 1995; Hampton, 2006), i.e., the closer a query is to its corresponding prototype, the more accurately it can be recognized. According to prototype theory, accurate emotion identification requires selecting the emotionally closest prototype for a given query and predicting the query's emotion based on emotional similarity. As shown in Figure 1(b), it involves three critical steps: (i) Constructing prototypes with accurate emotion representation. (ii) Choosing emotionally most similar prototype to serve as a reference. (iii) Predicting the query's emotion by assessing its emotional similarity with the chosen prototype. However, existing ICL exhibit limitations in all three of these steps:

(i) **Emotion Representation of Prototypes**. ICL tends to use single emotion labels to represent the emotions of demonstrations (i.e., prototypes), which oversimplifies the complexity of emotional states. For example, labeling the demonstration "This news makes me excited and anticipatory" merely as "excited" fails to capture the full range of emotions expressed.

(ii) **Prototype Selection**. ICL often selects prototypes based on semantic rather than emotional similarity, resulting in prototypes with limited emotional relevance. For example, the prototype "This news makes me anticipatory" provides little emotional insight for the query "This news makes me sad," even if they share semantic similarities.

(iii) **Emotion Prediction**. According to prototype theory, robust and nuanced emotion prediction requires focusing on emotions most similar to the prototype while eliminating in-

terference from dissimilar emotion types. In contrast, ICL treats all emotion categories as potential candidates, making it difficult to exclude irrelevant emotions, which leads to unstable and imprecise predictions.

To address the above limitations, we propose **E**motional **I**n-**C**ontext **L**earning (E-ICL) for fine-grained emotion recognition. E-ICL follows the three key steps of prototype theory to enhance emotion prediction: **First**, it applies a dynamic soft-labeling strategy to assign multiple emotion categories to demonstrations, constructing accurate emotional prototypes. **Second**, E-ICL choses emotionally similar examples as reference prototypes, rather than relying on semantic similarity. **Finally**, E-ICL employs an exclusion-based prediction strategy. It eliminates the interference of dissimilar emotions to prototypes, then guides LLMs to consider more similar emotions when predicting the query's emotion. **Importantly**, E-ICL achieves this through a plug-and-play emotion-capable auxiliary model, requiring no additional training. This design significantly enhances the method's flexibility and applicability.

We conduct extensive experiments on four fine-grained emotion recognition datasets: EDOS (Welivita et al., 2021), Empathetic-Dialogues (ED) (Rashkin et al., 2019), EmpatheticIntent (EI) (Welivita & Pu, 2020), and GoEmotions (GE) (Demszky et al., 2020). The experimental results demonstrate that compared to ICL, E-ICL guides LLMs to perceive fine-grained emotions more accurately with the assistance of different emotion-capable auxiliary models. Furthermore, more analyses show that E-ICL exhibits stable performance across different auxiliary models and LLMs. Notably, even when the performance of the auxiliary model is 10% lower than that of LLMs, the proposed method still enhances LLMs with a 4% higher performance than ICL on multiple datasets, indicating its stable advantage.

To sum up, our contributions are as follows: (i) To the best of our knowledge, we are the first to discover that In-Context Learning (ICL) aligns with prototype theory. This insight us to identify ICL's limitations in fine-grained emotion recognition tasks and propose E-ICL as a solution. (ii) We improve ICL's demonstration construction by developing strategies for retrieving emotionally similar examples and constructing dynamic soft labels, offering a new approach to demonstration construction. (iii) We introduce an exclusive emotion prediction strategy, enhancing the robustness and accuracy of emotion recognition. (iv) Experiments show that E-ICL exhibits an stable advantage in fine-grained emotion recognition across multiple datasets.

## 2 RELATED WORK

**Fine-grained Emotion Recognition**. The goal of the fine-grained emotion recognition task (FER) is to detect subtle emotion categories in the query (Liew & Turtle, 2016; Abdul-Mageed & Ungar, 2017). As emotions are primarily influenced by situational and cognitive factors (Gross et al., 2014; Siemer et al., 2007; Moors et al., 2013), existing works have mainly explored these two aspects and can be divided into situation-based models and cognition-based models. Situation-based Models mainly detect the subtle emotions implied in the query, without considering additional cognition information. These Models have explored word-level emotions (Li et al., 2020; Kim et al., 2021b; Yang et al., 2023b; Wang et al., 2024), mixed emotions (Majumder et al., 2020; Lin et al., 2019), and sentence-level emotions (Xie et al., 2019; Majumder et al., 2019; Ghosal et al., 2019). Cognition-based models mainly enhance emotions through additional cognitive factors. These models have explored aspects such as emotion causes (Gao et al., 2021; Kim et al., 2021a), and commonsense knowledge (Sabour et al., 2022; Li et al., 2020; Yang et al., 2024a;b). Both types of models have played an important role in the FER. However, these models are trained on specific datasets, limited by the corresponding data, and require certain computational resources and training time. In contrast, we explore the FER task through In-Context Learning, without consuming computational resources and training time.

**In-Context Learning**. In-Context Learning (ICL) improves LLMs' performance by learning from constructed demonstrations, circumventing the time and computational costs associated with fine-tuning. One part of ICL enhances LLMs by breaking down the reasoning steps of demonstrations into sub-steps and enabling LLMs to complete tasks by following these sub-steps (Wei et al., 2022; Hendrycks et al., 2021; Kazemi et al., 2022). This type of ICL has demonstrated satisfactory results in tasks such as arithmetic, commonsense, and symbolic reasoning (Rae et al., 2021). However, these methods involve a high cost of manual construction, and for some tasks, the objectives cannot

be directly decomposed into sub-process problems. Another part of ICL, i.e., retrieval-based ICL, mitigates this shortcoming by retrieving relevant demonstrations from training datasets. Retrieval-based ICL primarily retrieves demonstrations that are similar to the query in terms of words (Rubin et al., 2021; Agrawal et al., 2022; Luo et al., 2023), semantics (Li & Qiu, 2023; Liu et al., 2021; Yang et al., 2023a; Xiao et al., 2023; Liu et al., 2021), structures (Levy et al., 2022), or other relevant aspects (Fu et al., 2022; Gonen et al., 2022; Drozdov et al., 2022). Most of these methods rely on the semantics between the query and the demonstrations. Owing to the potential for semantic similarity to result in emotion misunderstanding issues, we propose an emotion-similarity-based retrieval approach and integrate it with an exclusionary emotion prediction mechanism to facilitate more accurate emotion prediction.

## 3 PRELIMINARIES

**Problem Formulation**. We formalize the fine-grained emotion recognition task as follows: given a query $Q$, the objective is to construct an effective prompt that guides a large language model (LLM) to accurately predict the fine-grained emotion category $C_Q$ expressed in $Q$. Here, $Q$ represents a sample in the test dataset, and $C_Q$ denotes one of $N_c$ fine-grained emotion categories $C$.

**Conceptual Clarification**. To elucidate our methodology more effectively, we clarify several key concepts. In our work, we equate prototypes with demonstrations. A demonstration comprises multiple example-label pairs, where examples are samples drawn from the training set. Their relationship is depicted as follows:

$$demonstrations = prototypes = [example\text{-}label_1, ..., , example\text{-}label_i] \qquad (1)$$

## 4 METHOD

**Overview**. Our proposed E-ICL is an in-context learning method that constructs and references emotionally accurate prototypes (i.e., demonstrations) for exclusionary emotion prediction, assisted by an auxiliary model. As shown in Figure 2, E-ICL consists of the following three steps: (i) **Prototype Construction** (Section 4.1). E-ICL employs a dynamic soft-label construction strategy to build prototypes (demonstrations) with accurate emotional representations. (ii) **Prototype Selection** (Section 4.2). It utilizes an emotion-similar example retrieval strategy to select prototypes that are emotionally closer to the query as references. (iii) **Emotion Prediction** (Section 4.3). It categorizes the query's emotions into those similar and dissimilar to the prototypes. It then prompts LLMs to prioritize similar emotions while excluding the interference of dissimilar emotions, thereby accurately predicting the emotion. Notably, the entire steps are facilitated by an emotion auxiliary model without requiring model training, thus enabling efficient emotion prediction while minimizing computational resources and time demands.

**Emotion Auxiliary Model**. We leverage emotion probabilities and emotion vectors generated from an emotion auxiliary model RoBERTa$_{large}^{emo}$ to enhance LLMs. Specifically, for an input $Input \in \{D_{test}, D_{train}\}$, we utilize RoBERTa$_{large}^{emo}$ to generate the corresponding emotion probabilities $P$ and emotion vector $V$.

$$P, V = RoBERTa_{large}^{emo}(Input), \qquad (2)$$

where $P \in R^{N_c}$ and $V \in R^{768}$. The emotion probabilities $P$ are used to construct dynamic soft labels and implement the exclusionary emotion strategy, while the generated vector $V$ is used to retrieve emotion-similar examples.

### 4.1 PROTOTYPE CONSTRUCTION

**Dynamic Soft Label Construction**. In line with prototype theory, we regard demonstrations as emotional prototypes. Previous approaches (Li & Qiu, 2023; Liu et al., 2021) only assign a single deterministic emotion label to the demonstrations. However, emotions are often complex and multi-faceted in linguistic expression (Larsen & McGraw, 2011; Crivelli & Fridlund, 2019; Trampe et al., 2015), thus such oversimplified labeling fails to capture this complexity, resulting in an inaccurate representation of emotional prototypes. To address this, we propose a dynamic soft label construction strategy. Specifically, we first employ the emotion auxiliary model to predict the emotions $e_{m_i}^s$

Figure 2: Overview of E-ICL. (1) E-ICL begins with a dynamic soft-label construction strategy to endow prototypes (demonstrations) with accurate emotional representations. (2) It then employs an emotion-similar retrieval strategy to select prototypes that are most emotionally relevant to the query. (3) Finally, it combines these prototypes with an exclusionary emotion prediction strategy to achieve robust and accurate predictions. Note that the entire steps, aided by the RoBERTa$_{large}^{emo}$ emotion auxiliary model, require no additional training.

and their corresponding probabilities $p_{m_i}^s$ for each sample $s_{m_i} \in D_{train}$; we then select the top $k_1$ emotions with the highest probabilities. Different samples have varied predicted emotions and probabilities, allowing for a more dynamic and nuanced emotion representation.

$$p_{m_i}^s, v_{m_i}^s = RoBERTa_{large}^{emo}(s_{m_i}), \tag{3}$$

$$e_i^s, p_i^s = Top_{k_1}(e_{m_i}^s, p_{m_i}^s), \tag{4}$$

where $p_{m_i}^s, p_i^s \in P, i \in [1, k_1], e_i^s \in C, m_i \in n_d$. $n_d$ is the number of samples in the training set. $Top_{k_1}$ is a ranking function that selects the top $k_2$ optimal emotions by their probabilities. $k_1$ is a hyperparameter.

Subsequently, We generate dynamic soft labels by combining predicted emotions with ground-truth labels, weighted by a hyperparameter $\alpha$, so we have:

$$p_i' = \begin{cases} 1 - \alpha \sum_{}^{k_1} p_j^s & \text{if } e_i = \text{Ground-Truth Label, and } i, j \in [1, k_1] \\ \alpha p_j^s & \text{Others, } j \in [1, k_1] \end{cases}. \tag{5}$$

By combining emotions $e_i$ with their corresponding probabilities $p_i'$, we obtain the dynamic soft label $l_{m_i}$ for the sample $s_{m_i}$. Incorporating the sample $s_{m_i}$ and its dynamic soft labels $l_{m_i}$, we derive the prototype $d_{m_i}$ with more accurate emotion representation.

$$l_{m_i} = (e_1, p_1') \oplus (e_2, p_2') \oplus ... \oplus (e_i, p_i'), \tag{6}$$

$$d_{m_i} = (s_{m_i}, l_{m_i}), \tag{7}$$

where $\oplus$ represents the concatenation operator, used to concatenate multiple label-probability pairs.

## 4.2 PROTOTYPE SELECTION

**Emotion-Similar Example Retrieval**. Although previous ICL approaches select semantically similar prototypes, they can still be emotionally incongruent or even contradictory to the query, which ultimately adversely affects prediction accuracy (Rosch & Mervis, 1975; Smith & Minda, 2002; Minda & Smith, 2001). To address this problem, we employ the emotion auxiliary model to retrieve emotion-similar examples. Specifically, we map the query $q_i \in D_{test}$ and a sample $s_{m_i} \in D_{train}$ into vectors using RoBERTa$_{large}^{emo}$, and calculate their similarity score $o_j$ via the cosine function.

We then rank the samples according to these similarity scores, and select the top $k_2$ highest-scoring samples as the emotion-similar examples $s_j$:

$$p_{q_i}, v_{q_i} = RoBERTa_{large}^{emo}(q_i), \tag{8}$$

$$o_{m_i} = Cosine(v_{q_i}, v_{m_i}^s), m_i \in n_d, \tag{9}$$

$$s_j = Top_{k_2}(o_1, o_2, ..., o_{m_i}), j \in [1, k_2], \tag{10}$$

where $v_{q_i}, v_{m_i}^s \in \mathbb{R}^{768}$, represent the emotion vector representations of the query $q_i$ and the sample $s_{m_i}$, respectively. $Top_{k_2}$ is a ranking function that selects the top $k_2$ optimal examples, with $k_2$ as a hyperparameter. Combining the selected examples $s_j$ and the constructed dynamic soft labels $l_i$, we obtain the prototypes $d_{q_i}$ as follows:

$$d_j = (s_j, l_i), \tag{11}$$

$$d_{q_i} = (d_1 \oplus d_2 \oplus ... \oplus d_{k_2}). \tag{12}$$

### 4.3 EMOTION PREDICTION

**Candidate Emotion Division**. Previous studies (Yang et al., 2023a; Xiao et al., 2023) attempt to predict emotions directly from a large set of categories. These approaches makes the choices made by LLMs abrupt, often resulting in suboptimal predictions. In contrast, when humans face complex choices, they tend to first eliminate unlikely options and then carefully consider the remaining possibilities (Tversky & Shafir, 1992; Dhar, 1997; Shafir et al., 1993; Payne et al., 1993). Inspired by this human decision-making process, we adopt an exclusionary emotion prediction strategy to enhance emotion prediction.

Our strategy begins by dividing the emotion categories into "possible" and "impossible" sets. To achieve this, we apply the emotion auxiliary model to predict the query's emotions. We then select the top $k_3$ emotions with the highest probabilities and consider them as possible emotions, which we place in the set $S_{pos}$. The remaining emotions are considered as impossible emotions and are placed in the set $S_{imp}$, so we have:

$$\widetilde{e}_i = Top_{k_3}(e_{q_i}, p_{q_i}), i \in [1, k_3], \tag{13}$$

$$\widetilde{e}_i \in S_{pos}, S_{imp} \cup S_{pos} = C, S_{imp} \cap S_{pos} = \emptyset, \tag{14}$$

where $\widetilde{e}_i, e_{q_i} \in C, p_{q_i} \in P$. $e_{q_i}$ and $p_{q_i}$ are the emotion categories and probabilities predicted by the auxiliary model for the query $q_i$, respectively. $\widetilde{e}_i$ represents the similar emotions. $Top_{k_3}$ is a selection function, and $k_3$ is a hyperparameter.

**Exclusionary Emotion Prediction**. Based on the above information, we predict fine-grained emotions in an exclusion-based strategy. Specifically, we prompt LLMs to comprehend the query and prototype, prioritizing emotions from the possible emotion set $S_{pos}$ before considering other emotions for prediction. The emotions in this set are similar to those expressed in the query, while the latter are similar to the prototype emotions. Indirectly, the emotions in the possible emotion set are likely to exhibit a high similarity to the prototype emotions. Subsequently, by combining the prototype and the highly similar candidate emotion set, we effectively eliminate interference from irrelevant emotions, thereby achieving more accurate and robust prediction of the query's emotion.

$$C_{q_i} = LLM(q_i, d_{q_i}, S_{pos}, S_{imp}). \tag{15}$$

## 5 EXPERIMENTS

**Emotion Auxiliary Model and Datasets**. To validate E-ICL, we conduct experiments using different emotion auxiliary models, RoBERTa$_{large}^{emo}$, on various datasets $D_{type}$, including EDOS (Welivita et al., 2021), Empathetic-Dialogues (ED) (Rashkin et al., 2019), EmpatheticIntent (EI) (Welivita & Pu, 2020), and GoEmotions (GE) (Demszky et al., 2020). Here, $emo \in \{EI, GE\}$, and $type \in \{EI, GE, ED, EDOS\}$. Note that our goal is to verify the performance of E-ICL without fine-tuning, so the auxiliary model used during inference should not have been fine-tuned on the respective dataset, i.e., $emo \neq type$. Simultaneously, the emotion categories predicted by the auxiliary model do not fully align with those of the datasets, rendering the exclusion strategy inapplicable. To address this issue, we adjust the datasets according to the emotion auxiliary model. For

Table 1: Results on the datasets when using the emotion auxiliary model RoBERTa$_{large}^{EI}$.

| LLM | Models | EDOS | | ED | | GE | |
|---|---|---|---|---|---|---|---|
| | | Acc | F1 | Acc | F1 | Acc | F1 |
| - | RoBERTa$_{large}^{EI}$ | 51.71 | 52.56 | 48.96 | 48.31 | 24.78 | 19.64 |
| Claude-haiku | Zero-Shot | 25.79 | 25.10 | 41.73 | 36.70 | 27.65 | 27.67 |
| | ICL | 36.79 | 38.61 | 49.47 | 47.01 | 36.6 | 33.04 |
| | E-ICL | **54.23** | **52.78** | **53.98** | **49.2** | **38.05** | **36.80** |
| ChatGPT-turbo | Zero-Shot | 34.6 | 34.14 | 36.4 | 29.82 | 33.17 | 29.70 |
| | ICL | 39.14 | 40.04 | 42.87 | 41.43 | 41.37 | 32.81 |
| | E-ICL | **54.45** | **54.37** | **51.56** | **49.32** | **46.1** | **37.19** |

example, for the RoBERTa$_{large}^{EI}$ auxiliary model (Welivita & Pu, 2020) and the GoEmotions dataset, we first identify the emotion categories they have in common. Then, we select data from GE that falls within these common emotion categories for experimentation. After this adjustment, the available datasets for the RoBERTa$_{large}^{EI}$ auxiliary model are GE, ED, and EDOS, with 19, 32, and 41 emotion categories, respectively. For the RoBERTa$_{large}^{GE}$ [1] auxiliary model, the available datasets are EI, ED, and EDOS, with 19, 17, and 19 emotion categories, respectively.

**Evaluation Metrics**. We utilize accuracy and macro-F1 for evaluating the methods, following the conventional approach. Accuracy measures the proportion of correctly predicted samples over the total samples. Macro-F1 is the harmonic mean of precision and recall, comprehensively considering both metrics. It accounts for the F1 score of each class and exhibits robustness against class imbalance.

**Baselines**. E-ICL leverages emotion auxiliary models to enhance the performance of large language models (LLMs) on fine-grained emotion recognition tasks. To validate the proposed method, we first employ the emotion auxiliary models RoBERTa$_{large}^{EI}$ and RoBERTa$_{large}^{GE}$ as baselines. RoBERTa$_{large}^{EI}$ and RoBERTa$_{large}^{GE}$ are RoBERTa models fine-tuned on the EI and GoEmotions emotion datasets, respectively. Secondly, we also select different large language models, namely ChatGPT3-turbo and Claude3-haiku, as baselines. On these LLMs, we construct zero-shot and semantic similarity-based ICL, denoted as Zero-Shot and ICL, respectively.

**Implementation Details**. Experimental details are provided in Appendix B.

## 6 RESULTS AND ANALYSIS

### 6.1 MAIN RESULTS

Table 1 presents the results using the RoBERTa$^{EI}large$ emotion auxiliary model. E-ICL significantly outperforms Zero-Shot and ICL methods, particularly on datasets with fine-grained emotions like EDOS and ED. This suggests E-ICL's superior ability to perceive and recognize nuanced emotions in queries. E-ICL also shows substantial improvements over the RoBERTa$^{EI}large$ model across all datasets. Notably, while the emotion auxiliary model's performance varies considerably between datasets, E-ICL maintains consistent performance, indicating greater robustness.

Table 2 shows results using the RoBERTa$_{large}^{GE}$ emotion auxiliary model. E-ICL significantly outperforms baselines on the EDOS dataset, demonstrating its advantage in fine-grained emotion recognition. ChatGPT-based E-ICL surpasses baselines on both ED and EDOS datasets, proving its effectiveness. However, Claude-based E-ICL doesn't show a clear advantage on these datasets. We attribute this to noise in the datasets, such as mixed English and Chinese characters in the EI dataset. This noise leads to inaccurate dynamic soft labels and eliminated emotion categories from the emotion auxiliary model. Additionally, Claude exhibits lower robustness compared to ChatGPT (detailed in Appendix C). Consequently, the less robust Claude-based E-ICL underperforms when faced with noisy data.

---

[1] https://huggingface.co/mrm8488/roberta-large-bne-finetuned-go_emotions-es

Table 2: Results on the datasets when using the emotion auxiliary model RoBERTa$_{large}^{GE}$.

| LLM | Models | EDOS | | ED | | EI | |
|---|---|---|---|---|---|---|---|
| | | Acc | F1 | Acc | F1 | Acc | F1 |
| - | RoBERTa$_{large}^{GE}$ | 43.25 | 43.63 | 40.64 | 40.07 | 41.24 | 41.67 |
| Claude-haiku | Zero-Shot | 42.87 | 37.83 | 53.22 | 51.81 | 53.64 | 50.57 |
| | ICL | 55.73 | 52.9 | 61.81 | 58.88 | 67.85 | 64.81 |
| | E-ICL | **62.16** | **57.74** | **62.08** | 57.99 | 66.16 | 62.04 |
| ChatGPT-turbo | Zero-Shot | 54.72 | 50.66 | 57.62 | 55.37 | 57.81 | 54.24 |
| | ICL | 56.99 | 54.33 | 58.18 | 56.27 | 61.49 | 55.28 |
| | E-ICL | **60.4** | **57.00** | **60.85** | **57.65** | 63.05 | 59.91 |

## 6.2 ANALYTICAL EXPERIMENTS

**Ablation Studies**. Figure 3 presents ablation studies using the RoBERTa$_{large}^{GE}$ emotion auxiliary model on the ED, EDOS, and GE datasets. Here, w/o DSL, w/o ESE, and w/o EEP represent the absence of dynamic soft label construction (Section 4.1), emotion-similar example retrieval (Section 4.2), and exclusionary emotion prediction strategies (Section 4.3), respectively. For the EDOS and ED datasets, removing any module decreases model performance, demonstrating that all three components contribute to accurate fine-grained emotion recognition. Conversely, on the GE dataset, removing DSL and EEP improves performance compared to the complete model. This is attributed to the dataset's significant noise, including mixed Chinese and English text and emoticons. Such noise leads the emotion auxiliary model to generate inaccurate dynamic soft labels and incorrectly eliminated emotions, resulting in suboptimal model performance.

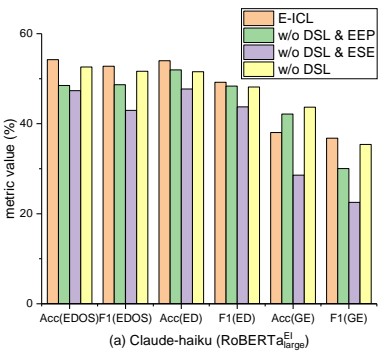
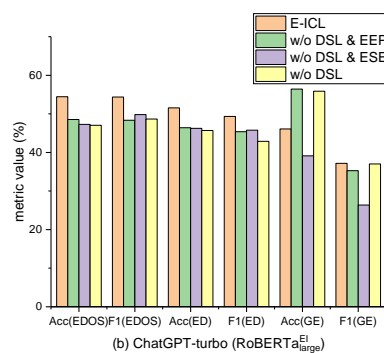

(a) Claude-haiku (RoBERTa$_{large}^{EI}$)     (b) ChatGPT-turbo (RoBERTa$_{large}^{EI}$)

Figure 3: Ablation experiments when using the emotionn auxiliar model RoBERTa$_{large}^{EI}$.

**Verifying the Contribution of Emotion Auxiliary Model to Dynamic Soft Labels**. We investigate the impact of parameter $\alpha$ on model performance. $\alpha$ determines the weight of emotion probabilities predicted by the emotion auxiliary model in dynamic soft labels. A higher $\alpha$ indicates greater influence from the auxiliary model. We examine two scenarios: one where the auxiliary model's emotion capability exceeds the LLM's, and another where it's weaker (details in Appendix D). Figure 4(a) illustrates the former case, while Figure 4(b) shows the latter. When assisted by a strong emotion auxiliary model, the emotion auxiliary models consistently enhance LLM performance, with minimal sensitivity to $\alpha$ variations. This is because only modulates emotion intensities ($p_i'$) in dynamic soft labels (equation 6), not emotion types ($e_i$). Since the emotion types already accurately represent the prototype of the example, they remain uninfluenced by $\alpha$. However, when paired with a weaker auxiliary model, performance initially increases and then decreases as $\alpha$ grows. This primarily occurs because the emotion types produced by the weaker auxiliary model are not highly accurate. A moderate consideration of the auxiliary model's judgments can lead to improved performance, whereas excessive reliance may be affected by the inaccurate judgments of the emotion auxiliary model.

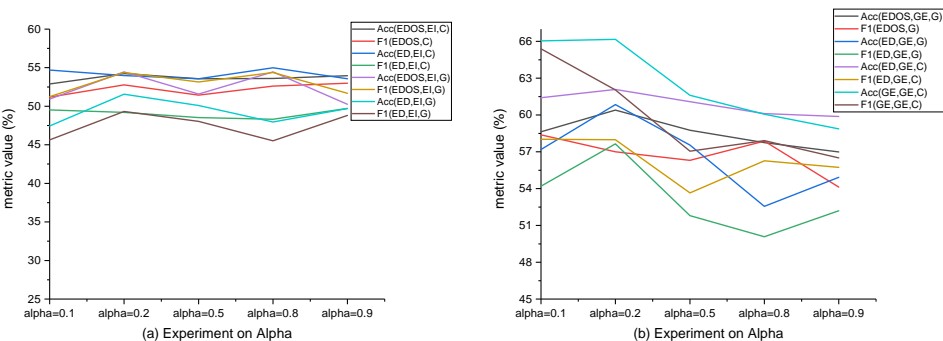

Figure 4: Results of E-ICL experiments across varying $\alpha$ values. Parenthetical elements indicate (dataset, emotion auxiliary model, LLM type). C and G denote Claude-haiku and ChatGPT, respectively.

**Impact of Dynamic Soft Label Quantity**. We assess the impact of varying the number of dynamic soft labels on E-ICL, with results shown in Figure 5. We categorize the experiments into two groups based on the emotional capability of the auxiliary models used, from high to low. Figure 5(a) depicts E-ICL results using auxiliary models with stronger emotional capability, while Figure 5(b) shows those with weaker capability. Comparing the two groups, we observe that as the number of soft labels increases: (1) The performance of the stronger capability group initially decreases, then improves. (2) The weaker capability group reaches a peak (or starts at a peak) before declining. These findings suggest that a moderate number of dynamic soft labels enhances emotion prediction in E-ICL. However, when emotion auxiliary models underperform compared to LLMs, increasing the number of emotion types for prototype representation reduces accuracy. This is due to the potential for misrepresentation when using an excessive number of emotion categories. For example, representing a prototype that inherently contains three emotions with ten emotion types can lead to inaccurate characterization. This misrepresentation ultimately degrades E-ICL performance.

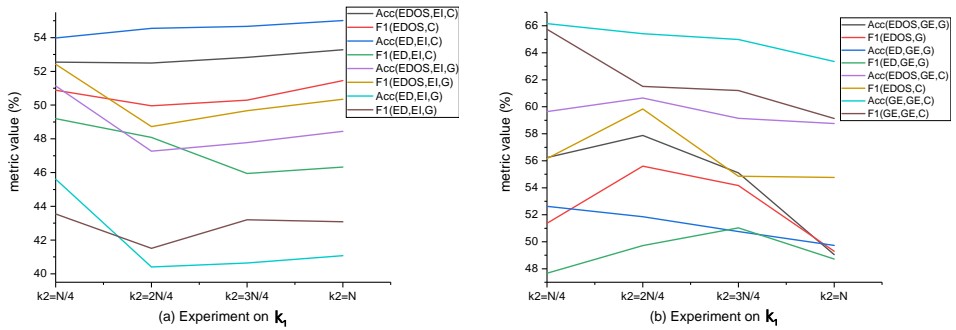

Figure 5: Experimental results of E-ICL based on different $k_1$, where N is the number of emotion categories in the dataset.

**Impact of Candidate Emotion Quantity**. We evaluate the impact of varying the number of candidate emotions ($k3$) on the exclusionary emotion prediction strategy. The experiments are divided into two groups based on the emotional capability of the auxiliary models: Figure 6(a) shows results from stronger emotion auxiliary models, while Figure 6(b) depicts those from weaker ones. The experimental results show that considering partial emotion categories instead of all categories during the prediction process leads to better performance on most datasets. This demonstrates the effectiveness of the exclusionary emotion prediction strategy.

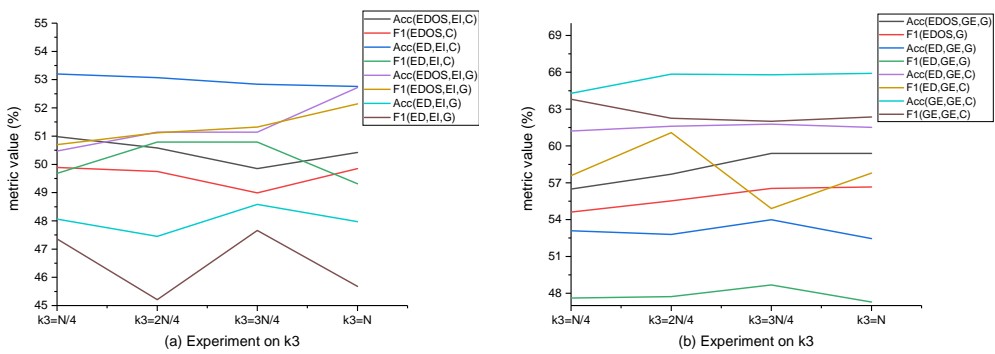

Figure 6: Experimental results of E-ICL based on different $k_3$, where N is the number of emotion categories in the dataset.

**More Analysis**. To further explore E-ICL, we conduct case studies, as detailed in Appendix E. Concurrently, we analyze the robustness of different LLMs to the introduced noise, as detailed in Appendix C.

## 7 DISCUSSION

**Conclusion**. This paper revealed that In-context learning (ICL) aligns with prototype theory for fine-grained emotion recognition. Based on prototype theory, we proposed Emotion In-Context Learning (E-ICL), which improved ICL using three steps: E-ICL first employs dynamic soft labeling to construct prototypes with more accurate emotion representations. It then retrieves examples with emotions similar to the query as reference prototypes. Finally, E-ICL utilizes an exclusionary emotion prediction strategy to eliminate interference from emotion types dissimilar to the prototypes, identifying the most similar emotions as candidates for predicting the query emotion. Experimental results and analysis have demonstrated that E-ICL achieves significant advantages on fine-grained recognition without requiring additional computational resources and training time.

**Limitations**. This paper has the following limitations: (i) Based on prior research (Xu et al., 2024; Liu et al., 2021), ICL is likely to conform to prototype theory across a broader range of tasks as well. (ii) While semantically similar example-label pairs to the query may not be the optimal choice for a wider range of tasks, we did not explore this aspect in our study. (iii) The exclusionary prediction strategy benefits ICL's accurate and robust judgments by avoiding interference from irrelevant categories, and it is likely applicable to multi-classification tasks. However, due to resource and time constraints, we do not further explore these limitations in this paper.

**Future Work**. In the future, we will investigate the following: (i) Explore whether ICL conforms to prototype theory in more tasks; (ii) Explore better methods for constructing example-label pairs; (iii) Study the applicability of the exclusionary prediction strategy in more tasks.

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

## A  PILOT EXPERIMENTS

We conduct experiments on four fine-grained emotion recognition datasets: EDOS, Empathetic-Dialogues, EmpatheticIntent, and GoEmotions. The results are shown in Figure 7. Taking Empathetic-Dialogues as an example, we first construct eight sets of examples, with each set containing five example-label pairs. Second, we treat the constructed example-label pairs as demonstrations and map them into vectors using RoBERTa$_{large}$. Subsequently, for each demonstration, we select 1216 queries based on similarity scores. We then assemble the demonstrations and queries as inputs to prompt the LLMs for emotion prediction. To eliminate interference from different LLMs, we perform experiments on both ChatGPT-turbo and Claude-haiku.

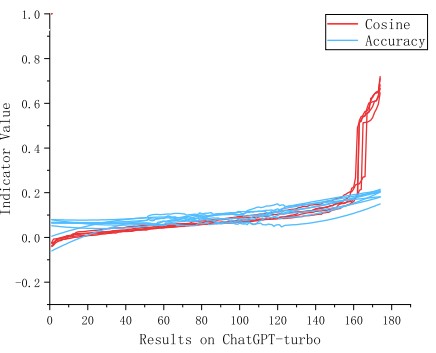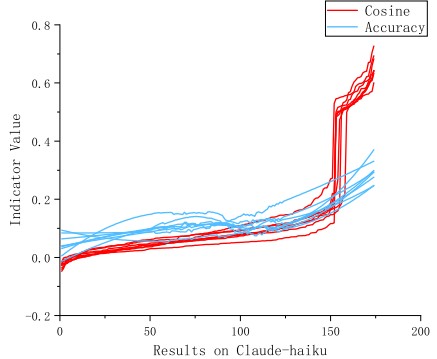

Figure 7: Results of the pilot experiment, where the red line represents the similarity between the example and the query, and the blue line denotes the ICL's emotion accuracy.

Table 3: Comparisons between RoBERTa$_{large}^{EI}$ and LLMs on different datasets, where positive values indicate RoBERTa$_{large}^{EI}$ outperforming LLMs, and negative values indicate the opposite.

| Comparison | EDOS | | ED | | GE | |
|---|---|---|---|---|---|---|
| | Acc | Macro-F1 | Acc | Macro-F1 | Acc | Macro-F1 |
| RoBERTa$_{large}^{EI}$ vs. Claude | 25.92 | 27.46 | 7.23 | 11.61 | -2.87 | -8.03 |
| RoBERTa$_{large}^{EI}$ vs. ChatGPT | 17.11 | 18.42 | 12.56 | 18.49 | -8.39 | -10.06 |

## B  IMPLEMENTATION DETAILS

In our experiments, we employ two emotion auxiliary models, RoBERTa$_{large}^{EI}$ and RoBERTa$_{large}^{GE}$, with the former being used for validation on the GE, ED, and EDOS datasets, while the latter is used for EI, ED, and EDOS. During the construction of the instance-label pairs, the example number is set to $k_2 = 5$. Meanwhile, the fusion weight for the soft labels is $\alpha = 0.2$. Additionally, the number of soft labels $k_1$ and the number of impossible emotions $k_1$ are influenced by variations in data, emotion auxiliary models, and LLMs. Consequently, we provide a detailed analysis and discussion of these factors in section 6.2.

## C  APPENDIX: ROBUSTNESS ANALYSIS OF LLMS

To validate the robustness of LLMs to E-ICL, we conduct the following experiments. First, we select Model RoBERTa$_{large}^{GE}$ as the emotion auxiliary model. Since this model performs relatively poorly on the respective EDOS, ED, and EI datasets, it will introduce more noise, which is beneficial for robustness experiments. Next, we select different numbers of candidate emotions to validate the robustness of Claude-haiku and ChatGPT-turbo. The experimental results are shown in Figure 3. The x-axis represents the number of candidate emotions, and the y-axis represents the metric values. As shown in Figure 8, as the number of candidate emotions (and noise) increases, the metric values of Claude-haiku fluctuate significantly, while those of ChatGPT-turbo remain stable within a certain range. This indicates that ChatGPT-turbo is more robust to the noise introduced by E-ICL.

## D  APPENDIX: GROUPED ANALYSIS OF EMOTION AUXILIARY MODELS

As shown in Tables 3 and 4, the emotion auxiliary models exhibit different performance across different datasets. Ignoring these differences and directly analyzing the experiments would lead to unreliable results. To investigate their impact, we divide the emotion auxiliary models into two groups: (a) those that significantly outperform LLMs, and (b) those that underperform LLMs. Specifically, we find that when using RoBERTa$_{large}^{EI}$ on the EDOS and ED datasets, its performance is significantly better than Claude-haiku and ChatGPT-turbo, while on the GE dataset, it underperforms compared to them. Therefore, we categorize the experiments based on RoBERTa$_{large}^{EI}$ and conducted on the EDOS and ED datasets as the (a) group experiments, while the experiments on the ED dataset are categorized as the (b) group experiments. Simultaneously, we adopt the same approach to divide the experiments based on RoBERTa$_{large}^{GE}$. Since RoBERTa$_{large}^{GE}$ does not significantly outperform LLMs on the EDOS, ED, and GE datasets, we categorize its experiments as the (b) group. When conducting parameter analysis experiments, due to the performance differences between the emotion auxiliary models and LLMs, the (a) group experiments and (b) group experiments exhibit different characteristics. This division of experiments better shows the impact of the emotion auxiliary models.

## E  APPENDIX: CASE STUDY

To demonstrate the advantages of E-ICL, we conduct a case study. The analysis results are shown in Table 5. The query of the 1st case expresses the emotion of "caring." The Zero-Shot method cannot accurately perceive this fine-grained emotion. In-context learning (ICL) predicts the query's emotion by retrieving and understanding semantically similar examples. However, the emotions of the semantically similar examples are diverse, such as "agreeing," "caring," and "grateful." Due to

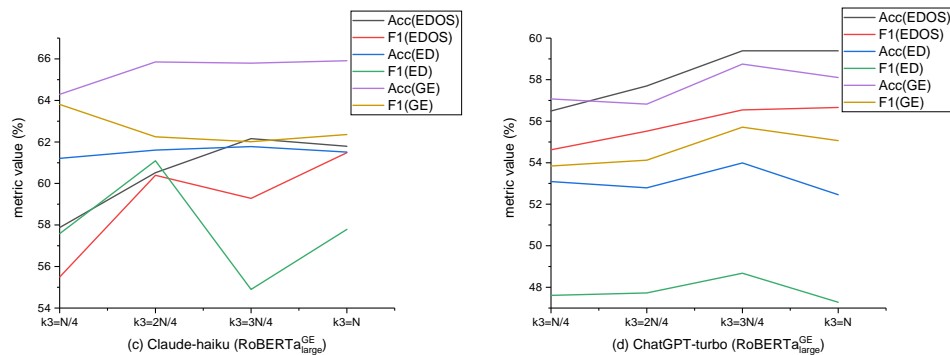

Figure 8: Experimental results of RoBERTa$_{large}^{GE}$ based E-ICL on different k3, where N is the number of emotion categories in the dataset.

Table 4: Comparison between RoBERTa$_{large}^{GE}$ and LLMs on different datasets, where positive values indicate RoBERTa$_{large}^{GE}$ outperforming LLMs, and negative values indicate the opposite.

| Comparison | EDOS | | ED | | EI | |
|---|---|---|---|---|---|---|
| | Acc | Macro-F1 | Acc | Macro-F1 | Acc | Macro-F1 |
| RoBERTa$_{large}^{GE}$ vs. Claude | 0.38 | 5.8 | -12.58 | -11.74 | -12.4 | -8.9 |
| RoBERTa$_{large}^{GE}$ vs. ChatGPT | -11.47 | -7.02 | -16.98 | -15.3 | -16.57 | -12.57 |

the difficulty in distinguishing among various emotions, ICL fails to accurately judge the emotion of the query, leading to an incorrect prediction of "encouraging." In contrast, E-ICL predicts the emotion of the query by retrieving and understanding examples with similar emotions, accurately predicting the query's emotion as "caring."

In the second case, the query expresses the emotion of "jealous." Similarly, the Zero-Shot method cannot accurately perceive this subtle emotion type. In the ICL method, the retrieved examples semantically similar to the query have diverse emotions, making it difficult for the LLM to accurately determine the emotion type based on these examples. In contrast, E-ICL retrieves three examples with similar emotions, enabling the LLM to make a more accurate prediction combined with these examples.

Table 5: Two case study of E-ICL and Benchmarks.

| Query | Hi , Tommy . I'm delivering a gift to Addie that's going to help get her back on track. **Emotion**: Caring | | |
|---|---|---|---|
| **Methods** | E-ICL | ICL | Zero-Shot |
| **Example 1** | When Mr Winters died they didn't have a replacement. I decided I'm going to rescue these poor kids. **Emotion**: Caring | It's your turn, mum .\I know! **Emotion**: Agreeing | – |
| **Example 2** | My grandmother's not doing so wel, so I took a year off from school to help her out. **Emotion**: Caring | Hey, there, Josey ! \We're going to Santo Rio! **Emotion**: Excited | – |
| **Example 3** | Pancho has a good heart. He feeds his little pet. **Emotion**: Caring | Good morning, Mr. Stark . \I brought you some homemade cookies. **Emotion**: Caring | – |
| **Example 4** | We came to the prisoners. **Emotion**: Caring | I have something for you. You'll be heard at Salamanca University . \ In a week! **Emotion**: Anticipating | – |
| **Example 5** | We built Graciela's casitas for abandoned women and children who needed a place to stay. **Emotion**: Sentimental | There you go, free sweets up for grabs. All you've got to do is get them out of the tube . \We placed everything needed within reach. **Emotion**: Grateful | – |
| **Prediction** | Caring | Encouraging | Hopeful |
| **Query** | I think she wants all the women around her to look fat. **Emotion**: Jealous | | |
| **Methods** | E-ICL | ICL | Zero-Shot |
| **Example 1** | All the women are around me in my office all day long, she's jealous over some foreign country I've never been to before. **Emotion**: Jealous | Everybody , listen ! Mr Anderson wants his team to play us . \A Japan-America All-Star game! **Emotion**: Anticipating | – |
| **Example 2** | Kimura . Jealousy makes me feel much younger. **Emotion**: Jealous | Natasha tries to get me out here once a week. **Emotion**: Annoyed | – |
| **Example 3** | Do you think my body is beautiful ? \I hate beautiful bodies . **Emotion**: Disgusted | Sanga ... Come and see her son do this . **Emotion**: Proud | – |
| **Example 4** | See those beauties ? \Know them ? Nope . \I wish! The one in red is hot. **Emotion**: Hopeful | According to my source, he's your gigolo . \I thought that men were the only ones who wanted to own women. But even young women have their personal toys. **Emotion**: Jealous | – |
| **Example 5** | They look like they're daring each other to move in. I hate that when a guy comes up to hit on you while his friends watch. **Emotion**: Jealous | He says he's waitin' for the presents. **Emotion**: Anticipating | – |
| **Prediction** | Jealous | Annoyed | Disgusted |

