# OpenReview forum: "Fine-Grained Emotion Recognition with In-Context Learning: A Prototype Theory Approach"
_ICLR.cc/2025/Conference — ICLR 2025 Conference Withdrawn Submission_

### Official Review · Reviewer_LwuJ · 2024-11-02

**Soundness:** 2
**Presentation:** 2
**Contribution:** 2
**Rating:** 3
**Confidence:** 5

**Summary:**

There are three main disadvantages according to In-context learning (ICL), including using a single emotion label leading to inaccurate emotion representation, referencing semantically similar but emotionally distant prototypes, and treating all emotion categories as candidates leading to interference and inaccurate predictions. This paper proposes an emotional context learning method (E-ICL) for fine-grained emotion recognition. Extensive experiments on fine-grained emotion datasets (EDOS, Empathetic-Dialogues, EmpatheticIntent, and GoEmotions) show that E-ICL significantly outperforms existing methods in emotion prediction performance.

**Strengths:**

N/A

**Weaknesses:**

1. In the description of Figure 1 (lines 88-90), the author does not introduce in detail which method is used to select the emotional prototype closest to the query. In addition, in lines 81-82, the author did not specifically explain why the existing methods would query irrelevant prototypes. It is recommended to add additional explanations to better reflect the limitations of the existing methods and the motivation of this article.
2. Some of the illustrations in Figure 1 are slightly rough. For example, the fonts of subtitles (a) and (b) are too large and inconsistent with other fonts.
3. In the selection of baseline models, although some common models and methods are selected as comparisons, there may be some other advanced emotional recognition methods that have not been compared. For example, some of the latest large language models and some fine-tuning methods are mentioned in the following literature.
[1] Liu Z, Yang K, Xie Q, et al. Emollms: A series of emotional large language models and annotation tools for comprehensive affective analysis[C]//Proceedings of the 30th ACM SIGKDD Conference on Knowledge Discovery and Data Mining. 2024: 5487-5496.
4. The authors need to give the recognition accuracy and F1 value of each method for each emotion category on the dataset to further illustrate the performance of the method. I need to clearly understand in which categories the performance of the proposed method is improved.
5. The paper simply explains the connection between ICL and prototype theory (lines 90-92) and proposes an improved method based on this. The theoretical support for its connection needs to be explained in the method or appendix.

**Questions:**

The important points listed in weakness 1-5.

---

### Official Review · Reviewer_T93f · 2024-11-03

**Soundness:** 1
**Presentation:** 1
**Contribution:** 1
**Rating:** 3
**Confidence:** 5

**Summary:**

The paper named “Fine-Grained Emotion Recognition with In-Context Learning: A Prototype Theory Approach” investigates the limitations of In-Context Learning (ICL) in fine-grained emotion recognition tasks. Using prototype theory, the authors identify key challenges: ICL’s reliance on single-emotion labels, selection of semantically rather than emotionally similar prototypes, and consideration of all emotion categories, which results in poor performance. To address these issues, they introduce Emotional In-Context Learning (E-ICL), a method that enhances prototype construction, selection, and prediction by incorporating a soft-labeling approach and exclusionary emotion prediction.

**Strengths:**

1. E-ICL effectively addresses ICL’s emotional recognition limitations by incorporating dynamic soft labels and emotionally similar prototype selection.

2. The approach shows a significant improvement in emotion recognition without additional model training, making it efficient and accessible.

**Weaknesses:**

1. There are some typos in the paper.
2. The paper primarily compares against ICL and zero-shot methods, which might not fully showcase E-ICL’s comparative strengths. There should be some methods with improved ICL prompts for comparison. It is not fair to directly compare zero-shot prompt with well-designed prompts.
3. The authors should include more LLMs to verify the effectiveness of the proposed E-ICL methods.

**Questions:**

NA

---

### Official Review · Reviewer_uvLi · 2024-11-05

**Soundness:** 2
**Presentation:** 1
**Contribution:** 1
**Rating:** 3
**Confidence:** 4

**Summary:**

The paper introduces Emotion In Context Learning (E-ICL), a method aimed to improve  the performance of few-shot LLMs on the task of fine-grained emotion detection. E-ICL proposes various improvements to traditional in-context learning methods: first, it leverages the interpretable nature emotion by recognizing that an example can express various emotion and various degrees. To this end, it proposes a soft-labeling strategy to assign multiple emotion categories to few-shot examples. Second the approach leverages embeddings of specialized emotion detection model to pick up “emotionally” similar demonstrations instead of relying only on similarity. Finally, E-ICL also incorporates an exclusion mechanism to focus only on demonstrations that are most likely to express the same emotion as the test example. The approach is tested on four fine-grained emotion detection datasets using two LLMs where it attains considerable performance improvements.

**Strengths:**

- The idea behind the paper is interesting. Using soft labels for prototypes seems promising and very fitting to emotion detection, an inherently ambiguous task.
- E-ICL considerably outperforms the traditional ICL.
- The paper carries out a lot of analyses to measure the effect of method hyperparameters: k1, k2, k3, alpha, as well as ablation study on various components of the method.

**Weaknesses:**

While the paper has some merit and the method is interesting, I believe there are significant flaws in both presentation and significance.

- The presentation of the method (i.e., Section 4) needs significant improvements. Notation is inconsistent and unintuitive, with various errors. I found it very hard to understand this section. For example, in equation 5 we subtract a vector from a scalar; there’s a sum over k1, but j is not properly defined; these probability distributions seem not to be probability distributions, i.e., they are not normalized. In L246 m_{i} \in n_{d} but in the very next sentence n_{d} is a scalar. There’s a notation of both p^{s}_{m_{i}} and p^{s}_{i}. Which one is it? In L205, V is presented as an emotion vector and it’s mentioned that V \in R^{768}. What is this 768? The reader may not be familiar with all embedding sizes for different types of models.
- The experimental setup is limited: only Claude-haiku and ChatGPT-turbo are not enough baselines to offer confidence that the method is widely applicable. Additionally, there are no baselines shown besides the RoBERTa.
- Most importantly, it is unclear to me why the RoBERTa model trained in-domain is not shown. In the method proposed here, the model has access to the training set and utilizes the GT (Eq 5). Therefore it’s critical to show this comparison where you train RoBERTa on the training data. Unfortunately, as shown in the papers introducing the datasets considered here, it seems that traditional language models such as BERT outperform E-ICL. Additionally E-ICL has significant inference-time costs that are not discussed here.
- The method is cumbersome: parameters such as k1, k2, k3, \alpha need to be tuned.

Missing citations:

Suresh et al., 2021 Not All Negatives are Equal: Label-Aware Contrastive Loss for Fine-grained Text Classification

**Questions:**

- Have you considered a variable number of demonstrations based on the context window (i.e., maxing out the input context length)? This could maybe remove the need for k2.
- Have you done a comparison of the inference costs ICL vs. E-ICL vs. RoBERTa.
- The prompt designs are not shown. How do the prompts look like?

---

### Note · Authors · 2025-01-17

I have read and agree with the venue's withdrawal policy on behalf of myself and my co-authors.